# Effect of different visual presentations on the public's comprehension of prognostic information using acute and chronic condition scenarios: two online randomised controlled trials

Eman Abukmail ![ORCID], Mina Bakhit ![ORCID], Mark Jones ![ORCID], Chris Del Mar, Tammy Hoffmann ![ORCID]

Prof. Chris Del Mar made invaluable contributions to this work, leaving a lasting impact on its development, with deep gratitude for his profound insights and expertise, we fondly remember him following his passing in 2022.

Institute for Evidence-Based Healthcare (IEBH), Faculty of Health Sciences and Medicine (HSM), Bond University, Robina, Queensland, Australia

**Correspondence to**
Dr Eman Abukmail;
eabukmai@bond.edu.au

## ABSTRACT

**Objectives** To assess the effectiveness of bar graph, pictograph and line graph compared with text-only, and to each other, for communicating prognosis to the public.

**Design** Two online four-arm parallel-group randomised controlled trials. Statistical significance was set at p<0.016 to allow for three-primary comparisons.

**Participants and setting** Two Australian samples were recruited from members registered at Dynata online survey company. In trial A: 470 participants were randomised to one of the four arms, 417 were included in the analysis. In trial B: 499 were randomised and 433 were analysed.

**Interventions** In each trial four visual presentations were tested: bar graph, pictograph, line graph and text-only. Trial A communicated prognostic information about an acute condition (acute otitis media) and trial B about a chronic condition (lateral epicondylitis). Both conditions are typically managed in primary care where 'wait and see' is a legitimate option.

**Main outcome** Comprehension of information (scored 0–6).

**Secondary outcomes** Decision intention, presentation satisfaction and preferences.

**Results** In both trials, the mean comprehension score was 3.7 for the text-only group. None of the visual presentations were superior to text-only. In trial A, the adjusted mean difference (MD) compared with text-only was: 0.19 (95% CI −0.16 to 0.55) for bar graph, 0.4 (0.04 to 0.76) for pictograph and 0.06 (−0.32 to 0.44) for line graph. In trial B, the adjusted MD was: 0.1 (−0.27 to 0.47) for bar graph), 0.38 (0.01 to 0.74) for pictograph and 0.1 (−0.27 to 0.48) for line graph. Pairwise comparisons between the three graphs showed all were clinically equivalent (95% CIs between −1.0 and 1.0). In both trials, bar graph was the most preferred presentation (chosen by 32.9% of trial A participants and 35.6% in trial B).

**Conclusions** Any of the four visual presentations tested may be suitable to use when discussing quantitative prognostic information.

**Trial registration number** Australian New Zealand Clinical Trials Registry (ACTRN12621001305819).

## STRENGTHS AND LIMITATIONS OF THIS STUDY

⇒ This study is the first to focus on communicating prognostic information about conditions managed in primary care and for which 'wait and see' is a legitimate option.
⇒ The study recruited a wide range of participants to ensure inclusivity and diversity of gender, background, age and educational level.
⇒ Baseline comprehension was not tested as it was not possible to ask specific comprehension questions without showing participants the interventions.
⇒ Trial participants did not need prior or current experience with the condition to be eligible, which may have influenced responses to some questions.

## INTRODUCTION

Providing patients with information on their health can encourage active participation in making decisions.[1] Having a conversation about prognosis can involve communicating quantitative data such as the likelihood of occurrence of an outcome, recurrence, or improvement over time, or the likely duration of the illness. Information about a condition's natural history (ie, the natural course of the condition without treatment) is also encompassed under the umbrella of prognostic information. Understanding prognostic information may help patients to have realistic expectations about the course of their condition and make well-informed decisions about subsequent steps.[2 3]

Communicating quantitative information can be challenging for both clinicians and patients[4] and most research has focused on how to communicate treatment benefits and harms. Several factors may contribute to patient's understanding of the information that is conveyed, including the format used and the framing of it.[5] We conducted

a systematic review[6] to evaluate the visual presentations (eg, a table, graph) that have been used to communicate prognostic information. The review identified only 11 studies and found no significant superior effect on comprehension for any of the visual presentations evaluated in the existing studies. It was unclear due to the lack of existing research if visual presentations would facilitate patients' understanding of the communicated information. Most studies presented prognostic information relevant to decision-making about undertaking screening or treatment for cancer.

Some decisions in primary care, for both acute and chronic conditions, may be enhanced by having prognostic information. Discussing prognostic information using visual presentations may improve patients' comprehension and facilitate decision-making that accommodates patients' values and preferences. The best ways of visually presenting prognostic information for conditions that are typically managed in primary care have rarely been explored. This study aimed to investigate, using conditions typically seen in primary care, whether graphs improve the comprehension of prognostic information compared with text-only, and if the type of graph matters. Within this study, we conducted two trials (A and B). In trial A, the scenario involved an acute condition (where a decision is needed within a few days), and trial B involved a chronic condition that requires a less urgent decision about how to manage it. Results from both trials informed the generalisability of our results for conditions managed in primary care.

## METHODS

Consolidated Standards of Reporting Trials (CONSORT) and The Checklist for Reporting Results of Internet E-Surveys (CHERRIES) were followed to report the results when applicable (see online supplemental additional file 1).

### Study design

The study consisted of two online four-arm parallel-group randomised controlled trials. Both trials had the same design and differed only in the condition used; trial A conveyed information about an acute condition (acute otitis media (AOM)) and trial B conveyed information about a chronic condition (lateral epicondylitis).

### Participants and recruitment

Participants were recruited between the 1st and 12th of October 2021 through an online research company, Dynata (www.dynata.com). Dynata recruited a national representative sample (for age, gender and education level) using an algorithm-based sampling tool and sent the surveys to existing registered members who had previously consented to complete online surveys. At the survey commencement, participants read study information and a statement about their right to withdraw from it at any time. To ensure validity and unique responses, Dynata used a captcha at the start of each survey as well as an IP-digital stamp for each participant. Participants were compensated with \$A5.90 by Dynata for survey completion. An invitation to the study was emailed to the participants individually using an automated router. Participants accessed the survey link via their Dynata dashboard. More information about Dynata Australian demographics, sampling and recruitment is provided in online supplemental additional file 1.

Participants were eligible if they lived in Australia, are 18 years old or older, and could read and understand English. Participants did not have to have the condition presented in the scenario. Participants can only participate in either trial A or B.

### Randomisation

In each trial, eligible participants were randomly allocated using a computer-generated sequence, generated by Dynata, to one of four groups (text-only, bar graph, pictograph, line graph).

### Trial scenarios and intervention groups

*Trial A:* the scenario conveyed information about the prognosis of children with AOM. AOM was chosen as it is one of the most common childhood infections,[7] with about 50%–85% of all children experiencing at least one episode.[8] Data came from a Cochrane systematic review.[8]

*Trial B:* the scenario conveyed information about the recovery of people with lateral epicondylitis who had completely recovered or not at two time points when they decided to watch and wait, have physiotherapy, or have a corticosteroid injection. We chose lateral epicondylitis as it is a chronic condition that has a few treatment options, including the option of waiting for the condition to spontaneously improve.[9]

Graphical details of bar graphs and pictographs (eg, symmetric distribution in pictograph, adding a scale, shape of the icon) were chosen based on best practice principles identified from relevant literature.[10–27] The interventions used are provided in online supplemental additional file 2.

### Patient and public involvement

We pilot tested the AOM materials in a convenience sample of day-care centre teachers (n=8) and the lateral epicondylitis materials with university administration staff (n=9). The feedback helped to iteratively refine minor details of the interventions and the clarity of some survey questions.

Before commencing trial recruitment, online piloting of all trial materials was conducted with approximately 10% of the sample size needed for the full trial (trial A, n=47; trial B, n=41). Participants were recruited by Dynata and not invited to subsequently participate in the trials. Piloting data were also used to adjust the sample size calculation and test for any technical problems. These data were not included in the trial results.

## Data collection and outcome measurement

Our primary outcome was comprehension of the information presented, measured using six questions based on previous studies[12 16 23 27] (see online supplemental additional file 1 for questions). The comprehension questions were developed to assess different understanding skills, for example, the ability to extract numbers from the information provided (eg, at 10–12 days, out of 100 children who took antibiotics, how many had pain?), the ability to use the numbers to compare the two options (eg, which group was less likely to experience pain at about 4–7 days?) and the ability to make use of the information (eg, in the week or so after a middle ear infection starts, the majority of children will NOT be in pain, regardless of whether they do or do not take antibiotics). Therefore, we were interested to assess overall comprehension differences across groups rather than differences in comprehension of individual questions. Secondary outcomes were decision intention, satisfaction with the presented format (before revealing all formats), and format preference (after revealing all formats) and measured with a combination of ranking questions, Likert scales and open-ended questions (online supplemental additional file 1). All outcomes were measured once (immediately postintervention), except for decision intention which was measured preintervention and postintervention (see online supplemental additional file 2 for the results of individual questions).

Participants completed a three-part, 28-question structured survey developed for the trial, based on surveys from relevant research. Part 1: five demographic questions, a baseline decision intention question, a question on previous experience with the condition, the Medical Maximiser-Minimiser Scale (assesses patients' preferences for aggressive vs more passive approaches to healthcare),[28] and a validated Subjective Numeracy Scale.[29 30] Part 2: each group was presented with information according to their allocated group (eg, text-only, bar graph, pictograph or line graph). This was followed by six comprehension questions, two questions about decision intention and two about satisfaction with the intervention. Part 3: all four intervention presentations (text-only, bar graph, pictograph and line graph) were revealed to participants and visual presentation preference was measured using three questions. Participants rated their previous graphic experience on a 1–5 scale. Health literacy level was measured using the Newest Vital Sign scale.[31]

The multiple-choice options for the comprehension and decision intention questions were randomised to minimise order bias. Participants were able to see the randomised intervention while answering the questions in part 2, but once they moved to the next page, they were unable to go back to change their responses. This was to ensure that responses to comprehension questions were not influenced by the other interventions that were revealed in part 3. Data are available at OSF | Effect of different visual presentations on the public's comprehension of prognostic information using acute and chronic condition scenarios: two online randomised controlled trials.[32]

## Sample size

In the online piloting, the SD for the primary outcome was, at most, 1.66 units. Assuming 90% power, 0.016% level of significance (adjusted to account for three primary comparisons), the difference between groups of 1 unit, and up to 20% missing data, 97 participants were calculated as required in each group (a total of 388 participants) in each trial.

## Data analysis

The primary analysis tested whether comprehension was different between each graphic presentation and the text-only presentation (three comparisons). Linear regression was used and mean differences (MDs) with 95% CIs and p values calculated. Adjusted analysis was performed to account for any potential important baseline imbalances using linear regression, with adjustment for age group, education level and health literacy level. Statistical significance was set at p<0.016 to allow for three primary comparisons.

To assess the equivalence of the three graphic presentations, 95% CIs for the MD between groups were estimated for each of the three pairwise comparisons using linear regression. Equivalence was to be concluded if the 95% CIs were between −1.0 and 1.0. We were guided by a previous study that used 10% as a clinically important difference for a similar comprehension outcome measure.[33] Univariable linear regression models were constructed to estimate the association between comprehension as the dependent variable and the independent variables (including health literacy, numeracy and education level). The measure of association reported is the $R^2$ value. SAS OnDemand for Academics 2014 was used to analyse the data.

For the four open-ended questions (eg, the most important reason for participant's decision to choose one of the management options), two authors (EA, MB) independently categorised the responses. Another author (TH) was consulted when there was disagreement. Each response could be categorised into one or more categories (to a maximum of three). Decision intention and visual presentation satisfaction and preferences were descriptively reported as number and percentage of participants.

## Postprotocol analysis and modifications from protocol

It was not feasible to analyse the data blinded to group allocation because we needed to identify the control group (text-only) so that the appropriate comparisons could be made.

We estimated the association between comprehension as the dependent variable and time to finish the comprehension questions and previous experience with graphs as independent variables (results are provided in online supplemental additional file 2).

As part of the review process, we conducted statistical analyses for the secondary outcomes of decision intention and satisfaction. Change in decision intention from preintervention to postintervention was compared between groups using a multinomial logistic regression model with cluster robust standard errors specified to account for the pre/post repeated measures on each participant. The dependent variable was decision intention category, and the independent variables were group, time (pre/post) and the interaction between group and time. A joint test was used to test for evidence of an interaction between group and time with statistical significance set at $p < 0.05$.

Analysis of variance was used to test for differences in the satisfaction outcomes between intervention groups with statistical significance set to $p < 0.05$.

## RESULTS

Figure 1 shows the CONSORT flow diagram of participants in both trials. Table 1 presents the baseline demographic results. There were no baseline differences among the groups in either trial, except for health literacy which was slightly higher in the pictograph group in trial A.

### Trial A
#### Comprehension
Figure 2A shows the mean comprehension score for each group. For the percentage of participants, in each group, who correctly answered the comprehension questions, see online supplemental additional file 2. The total mean score across all questions was 3.71 (SD 1.63), 3.79 (SD 1.54), 3.81 (SD 1.53) and 4.25 (SD 1.40) for text-only,

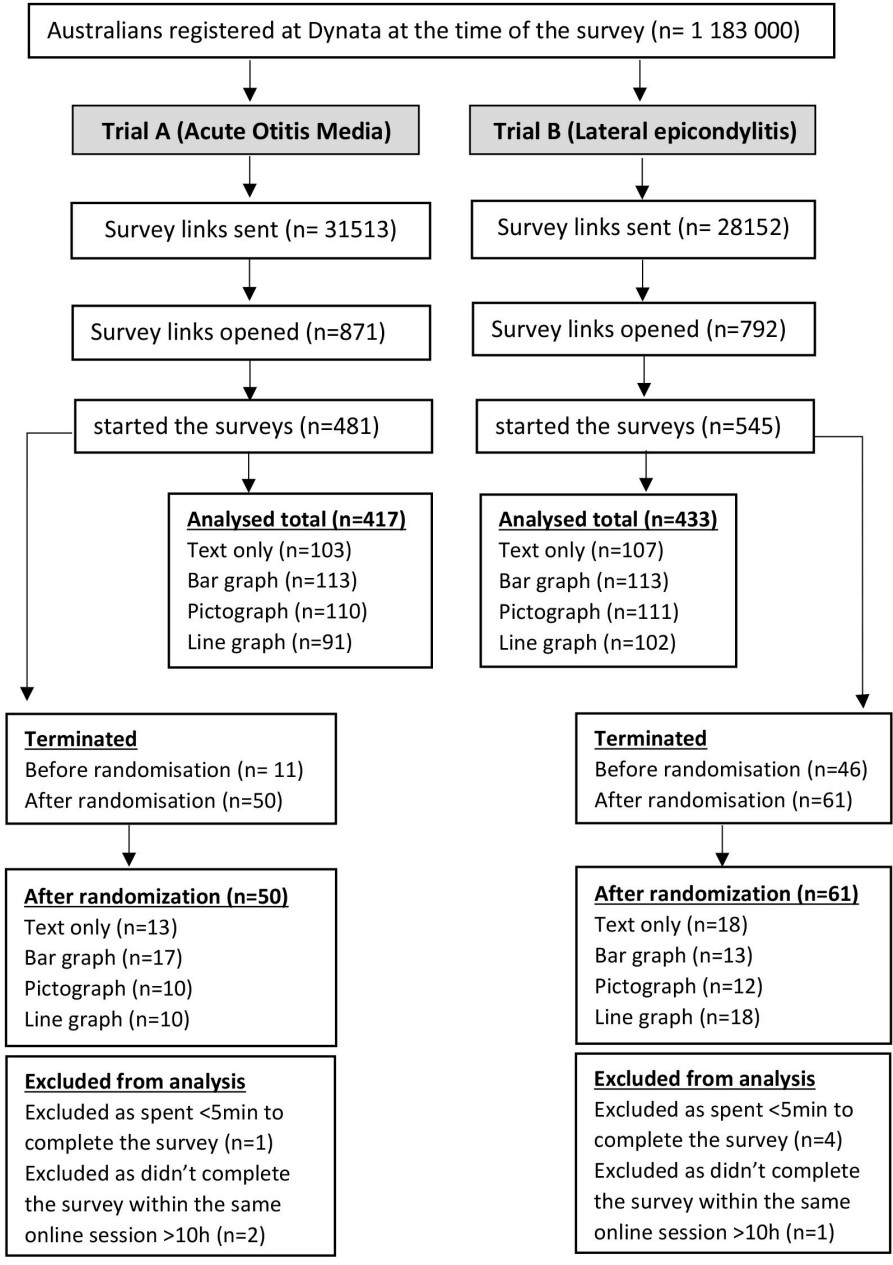

**Figure 1** Consolidated Standards of Reporting Trials flow diagrams of participants in trial A and trial B.

**Table 1** Participants' characteristics

| | Trial A | | | | Trial B | | | |
|---|---|---|---|---|---|---|---|---|
| | Text only (n=103) | Bar graph (n=113) | Pictograph (n=110) | Line graph (n=91) | Text only (n=107) | Bar graph (n=113) | Pictograph (n=111) | Line graph (n=102) |
| **Age (years) n (%)** | | | | | | | | |
| 18–24 | 5 (4.9) | 7 (6.2) | 7 (6.4) | 4 (4.4) | 5 (4.7) | 6 (5.3) | 5 (4.5) | 10 (9.8) |
| 25–34 | 15 (14.6) | 17 (15.0) | 18 (16.4) | 8 (8.8) | 17 (15.9) | 22 (19.5) | 12 (10.8) | 15 (14.7) |
| 35–44 | 14 (13.6) | 19 (16.8) | 14 (12.7) | 18 (19.8) | 21 (19.6) | 17 (15.0) | 25 (22.5) | 16 (15.7) |
| 45–54 | 25 (24.3) | 23 (20.4) | 30 (27.3) | 8 (8.8) | 20 (18.7) | 23 (20.4) | 19 (17.1) | 14 (13.7) |
| 55–64 | 26 (25.2) | 24 (21.2) | 23 (20.9) | 28 (30.8) | 24 (22.4) | 24 (21.2) | 28 (25.2) | 21 (20.6) |
| 65+ | 18 (17.5) | 23 (20.4) | 18 (16.4) | 25 (27.5) | 20 (18.7) | 21 (18.6) | 21 (18.9) | 26 (25.5) |
| Prefer not to say | 0 (0.0) | 0 (0.0) | 0 (0.0) | 0 (0.0) | 0 (0.0) | 0 (0.0) | 1 (0.9) | 0 (0.0) |
| **Gender n (%)** | | | | | | | | |
| Male | 48 (46.6) | 53 (46.9) | 49 (44.6) | 47 (51.7) | 52 (48.6) | 44 (38.9) | 46 (41.4) | 57 (55.9) |
| Female | 55 (53.4) | 60 (53.1) | 60 (54.6) | 44 (48.4) | 55 (51.4) | 69 (61.1) | 64 (57.7) | 45 (44.1) |
| Others | 0 (0.0) | 0 (0.0) | 1 (0.9) | 0 (0.0) | 0 (0.0) | 0 (0.0) | 1 (0.9) | 0 (0.0) |
| **State n (%)** | | | | | | | | |
| QLD | 26 (25.2) | 31 (27.4) | 18 (16.4) | 24 (26.4) | 23 (21.5) | 20 (17.7) | 24 (21.6) | 22 (21.6) |
| NSW | 31 (30.1) | 34 (30.1) | 38 (34.6) | 33 (36.3) | 32 (29.9) | 39 (34.5) | 41 (36.9) | 33 (32.4) |
| ACT | 0 (0.0) | 2 (1.8) | 3 (2.7) | 1 (1.1) | 2 (1.9) | 3 (2.7) | 1 (0.9) | 2 (2.0) |
| NT | 1 (1.0) | 0 (0.0) | 0 (0.0) | 0 (0.0) | 1 (0.9) | 0 (0.0) | 1 (0.9) | 1 (1.0) |
| WA | 12 (11.7) | 7 (6.2) | 17 (15.5) | 4 (4.4) | 8 (7.5) | 17 (15.0) | 11 (9.9) | 10 (9.8) |
| VIC | 22 (21.4) | 31 (27.4) | 23 (20.9) | 23 (25.3) | 28 (26.2) | 28 (24.8) | 21 (18.9) | 22 (21.6) |
| TAS | 2 (1.9) | 2 (1.8) | 1 (1.0) | 0 (0.0) | 1 (0.9) | 2 (1.8) | 2 (1.8) | 3 (2.9) |
| SA | 9 (8.7) | 6 (5.3) | 10 (10.0) | 6 (6.6) | 12 (11.2) | 4 (3.5) | 10 (9.0) | 9 (8.8) |
| **Educational level n (%)** | | | | | | | | |
| High school Y10 | 13 (12.6) | 19 (16.8) | 12 (10.9) | 10 (11.0) | 12 (11.2) | 9 (8.0) | 11 (9.9) | 12 (11.8) |
| High school Y12 | 11 (10.7) | 20 (17.7) | 25 (22.7) | 16 (17.6) | 14 (13.1) | 17 (15.0) | 19 (17.1) | 13 (12.8) |
| Certificate I–IV, diploma or apprenticeship | 31 (30.1) | 32 (28.3) | 32 (29.1) | 25 (27.5) | 39 (36.5) | 34 (30.1) | 35 (31.5) | 33 (32.4) |
| Undergraduate degree‡ | 32 (31.1) | 29 (25.7) | 28 (25.5) | 28 (30.8) | 29 (27.1) | 37 (32.7) | 33 (29.7) | 27 (26.5) |
| Postgraduate degree‡ | 16 (15.5) | 13 (11.5) | 13 (11.8) | 12 (13.2) | 13 (12.2) | 16 (14.2) | 13 (11.7) | 17 (16.7) |
| SNS mean‡ (SD) | 4.2 (0.9) | 4.2 (1.1) | 4.2 (1.0) | 4.2 (1.0) | 4.1 (1.0) | 4.2 (1.1) | 4.2 (1.1) | 4.2 (1.0) |
| SNS ability | 4.3 (1.2) | 4.3 (1.4) | 4.3 (1.2) | 4.3 (1.2) | 4.1 (1.3) | 4.1 (1.4) | 4.1 (1.5) | 4.3 (1.3) |
| SNS preferences | 4.2 (0.8) | 4.1 (1.1) | 4.2 (0.9) | 4.1 (1.0) | 4.1 (0.9) | 4.2 (1.0) | 4.2 (1.1) | 4.1 (1.0) |
| **Health literacy n (%)** | | | | | | | | |
| 4–6 adequate | 60 (58.3) | 77 (68.1) | 86 (78.2) | 59 (64.8) | 68 (63.55) | 76 (67.26) | 80 (72.07) | 67 (65.69) |
| 2–3 possibly limited | 38 (36.9) | 25 (22.1) | 19 (17.3) | 24 (26.4) | 24 (22.43) | 23 (20.35) | 16 (14.41) | 26 (25.49) |
| 0–1 highly limited | 5 (4.9) | 11 (9.7) | 5 (4.6) | 8 (8.8) | 15 (14.02) | 14 (12.39) | 15 (13.51) | 9 (8.82) |
| **MMM scale‡ n (%)** | | | | | | | | |
| <4 minimiser | 54 (52.4) | 65 (57.5) | 61 (55.5) | 52 (57.1) | 63 (58.9) | 65 (57.5) | 62 (55.9) | 47 (46.1) |
| =4 neutral | 31 (30.1) | 32 (28.3) | 31 (28.2) | 21 (23.1) | 31 (29.0) | 27 (23.9) | 28 (25.2) | 34 (33.3) |
| >4 maximiser | 18 (17.5) | 16 (14.2) | 18 (16.4) | 18 (19.8) | 13 (12.2) | 21 (18.6) | 21 (18.9) | 21 (20.6) |
| **Experienced with the condition n (%)** | 47 (45.6) | 48 (42.5) | 50 (45.5) | 46 (50.6) | 32 (29.9) | 46 (40.7) | 39 (35.1) | 26 (25.5) |
| **Time (min) to finish comprehension Qs Median (IQR)** | 3.5 (2.6–5.3) | 3.5 (2.5–5.1) | 3.5 (2.4–5.4) | 3.3 (2.3–4.8) | 3.4 (2.2–4.6) | 3.2 (2.1–4.2) | 3.2 (2.6–4.6) | 3.1 (2.2–4.6) |
| **Speaking English at home n (%)** | 98 (95.15) | 105 (92.92) | 101 (91.82) | 86 (94.51) | 100 (93.5) | 101 (89.4) | 103 (92.8) | 94 (92.2) |

| | Trial A | | | | Trial B | | | |
|---|---|---|---|---|---|---|---|---|
| | Text only (n=103) | Bar graph (n=113) | Pictograph (n=110) | Line graph (n=91) | Text only (n=107) | Bar graph (n=113) | Pictograph (n=111) | Line graph (n=102) |

*Bachelor's degree or equivalent.
†Masters or doctoral degree, or equivalent.
‡SNS: is a self-report measure of numeracy and scored as the average rating of eight items (four items assessed ability and four items assessed preferences) marked (1–6).
§Medical Maximiser-Minimiser Scale (MMMS): assesses patients' preferences for aggressive versus more passive approaches to healthcare, medical maximisers (seek healthcare even for minor problems), medical minimisers: (avoid medical intervention unless it is necessary).
¶Percentages were reported to one decimal.
SNS, Subjective Numeracy Scale.

bar graph, line graph and pictograph groups, respectively. Only pictograph group participants had statistically significantly superior comprehension to the text-only group participants, with an MD of 0.54 (95% CI 0.13 to 0.95, p value 0.011).

There was no significant difference between the bar graph group and the text-only group or between the line graph group and the text-only group (table 2). After adjustment for age group, education level and health literacy: pictograph was not statistically different compared with text-only (MD 0.40, 95% CI 0.04 to 0.76, p value 0.031). Although comprehension was higher in participants in the pictograph group than those in the other two graph groups, the 95% CIs are within the

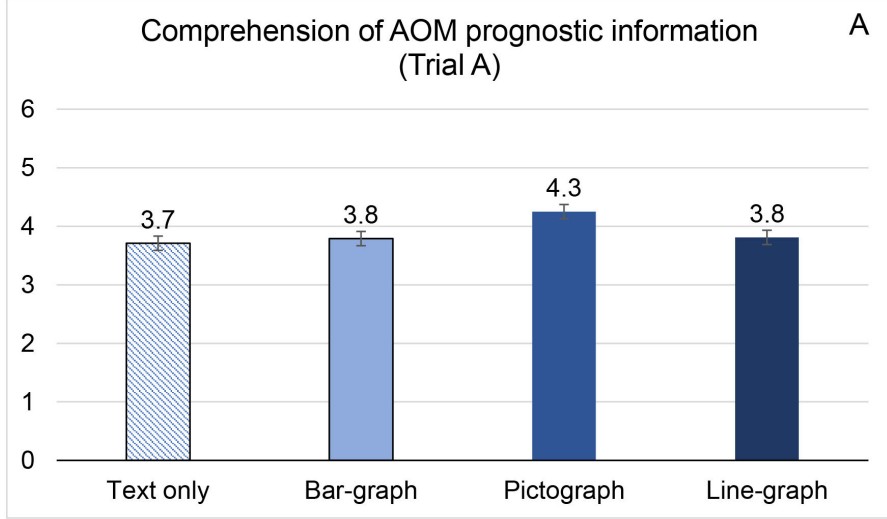

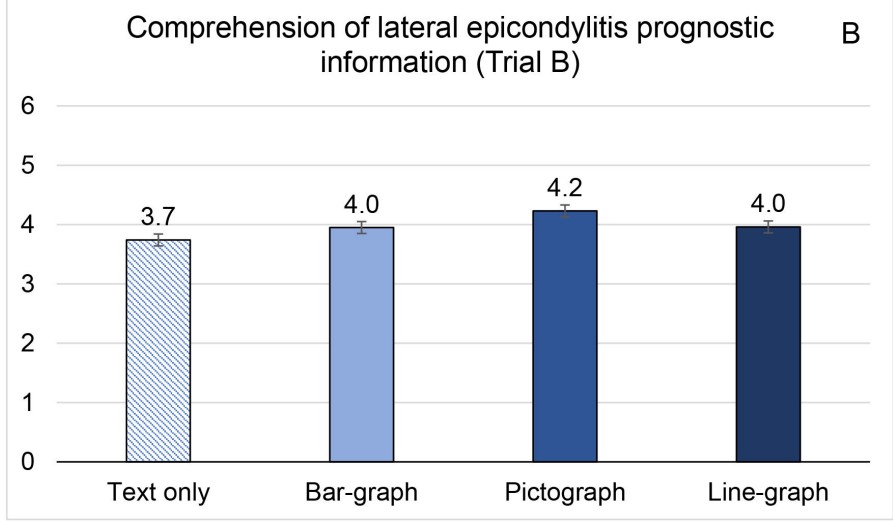

**Figure 2** Mean comprehension score of prognostic information per group in trial A (AOM) (A) and trial B (lateral epicondylitis) (B). Comprehension questions are provided in online supplemental additional file 1. Trial A and B comprehension questions are different. AOM, acute otitis media.

**Table 2** Superiority comparisons of mean comprehension scores, decision intention, satisfaction with the presentation, visual presentation preference in all intervention groups, in trial A and trial B

| | Trial A | | | | Trial B | | | |
|---|---|---|---|---|---|---|---|---|
| | Text only (n=103) | Bar graph (n=113) | Pictograph (n=110) | Line graph (n=91) | Text only (n=107) | Bar graph (n=113) | Pictograph (n=111) | Line graph (n=102) |
| **Superiority comparison (not adjusted)** | | | | | | | | |
| Mean (SD) | 3.71 (1.63) | 3.79 (1.54) | 4.25 (1.40) | 3.81 (1.53) | 3.74 (1.60) | 3.95 (1.59) | 4.23 (1.59) | 3.96 (1.89) |
| MD | Ref | 0.08 | 0.54 | 0.1 | Ref | 0.21 | 0.49 | 0.22 |
| 95% CI | Ref | −0.33 to 0.49 | 0.13 to 0.95 | −0.33 to 0.54 | Ref | −0.23 to 0.65 | 0.04 to 0.93 | −0.23 to 0.68 |
| P value | Ref | 0.70 | 0.011 | 0.63 | Ref | 0.35 | 0.032 | 0.34 |
| **Superiority comparison (adjusted for age group, level of education and categorised health literacy)** | | | | | | | | |
| MD* | Ref | 0.19 | 0.4 | 0.06 | Ref | 0.1 | 0.38 | 0.1 |
| 95% CI | Ref | −0.16 to 0.55 | 0.04 to 0.76 | −0.32 to 0.44 | Ref | −0.27 to 0.47 | 0.01 to 0.74 | −0.27 to 0.48 |
| P value | Ref | 0.29 | 0.031 | 0.76 | Ref | 0.59 | 0.046 | 0.59 |
| **Decision intention n (%) before the intervention** | | | | | | | | |
| Take antibiotics | 68 (66.0) | 73 (64.6) | 69 (62.7) | 62 (68.1) | | | | |
| Not to take antibiotics | 18 (17.5) | 17 (15.0) | 18 (16.4) | 17 (18.7) | | | | |
| Unsure | 17 (16.5) | 23 (20.4) | 23 (20.9) | 12 (13.2) | 10 (9.4) | 11 (9.7) | 12 (10.8) | 12 (11.8) |
| To wait and watch | | | | | 58 (54.2) | 63 (55.8) | 61 (55.0) | 45 (44.1) |
| To have physiotherapy | | | | | 27 (25.2) | 28 (24.8) | 28 (25.2) | 29 (28.4) |
| To have corticosteroid | | | | | 12 (11.2) | 11 (9.7) | 10 (9.0) | 16 (15.7) |
| **Decision intention n (%) after the intervention** | | | | | | | | |
| Take antibiotics | 75 (72.8) | 73 (64.6) | 87 (79.1) | 67 (73.6) | | | | |
| Not to take antibiotics | 13 (12.6) | 22 (19.5) | 14 (12.7) | 12 (13.2) | | | | |
| Unsure | 15 (14.6) | 18 (15.9) | 9 (8.2) | 12 (13.2) | 5 (4.7) | 7 (6.2) | 5 (4.5) | 7 (6.9) |
| To wait and watch | | | | | 39 (36.4) | 47 (41.6) | 47 (42.3) | 34 (33.3) |
| To have physiotherapy | | | | | 45 (42.1) | 43 (38.1) | 47 (42.3) | 49 (48.0) |
| To have corticosteroid | | | | | 18 (16.8) | 16 (14.2) | 12 (10.8) | 12 (11.8) |
| **Satisfaction mean (SD)** | | | | | | | | |
| How easy (1–10)† | 7.2 (2.2) | 7.3 (2.5) | 7.7 (2.0) | 7.6 (2.2) | 7.8 (1.8) | 7.8 (2.2) | 8.3 (1.8) | 7.5 (2.6) |
| How satisfied (1–10)‡ | 7.0 (2.2) | 7.4 (2.4) | 7.7 (2.0) | 7.6 (2.2) | 7.8 (1.9) | 8.0 (2.0) | 8.3 (1.9) | 7.7 (2.4) |
| **Visual presentation preference—most preferred (first) N (%)** | | | | | | | | |
| Text only | 40 (38.8) | 38 (33.6) | 25 (22.7) | 21 (23.1) | 43 (40.2) | 40 (35.4) | 27 (24.3) | 34 (33.3) |
| Bar graph | 31 (30.1) | 43 (38.1) | 34 (30.9) | 29 (31.9) | 40 (37.4) | 56 (49.6) | 28 (25.2) | 30 (29.4) |
| Pictograph | 9 (8.7) | 8 (7.1) | 34 (30.9) | 7 (7.7) | 12 (11.2) | 8 (7.1) | 45 (40.5) | 3 (2.9) |
| Line graph | 23 (22.3) | 24 (21.2) | 17 (15.5) | 34 (37.4) | 12 (11.2) | 9 (8.0) | 11 (9.9) | 35 (34.3) |
| **Visual presentation preference—least preferred (fourth) N (%)** | | | | | | | | |
| Text only | 29 (28.2) | 25 (22.1) | 47 (42.7) | 29 (31.9) | 20 (18.7) | 30 (26.6) | 41 (36.9) | 32 (31.4) |
| Bar graph | 10 (9.7) | 12 (10.6) | 12 (10.9) | 15 (16.5) | 6 (5.6) | 8 (7.1) | 9 (8.1) | 7 (6.9) |
| Pictograph | 43 (41.8) | 56 (49.6) | 25 (22.7) | 34 (37.4) | 41 (38.3) | 34 (30.1) | 11 (9.9) | 45 (44.1) |
| Line graph | 21 (20.4) | 20 (17.7) | 26 (23.6) | 13 (14.3) | 40 (37.4) | 41 (36.3) | 50 (45.1) | 18 (17.7) |
| **Previous graph experience§ mean (SD)** | | | | | | | | |
| Bar graph (1–5) | 3.3 (1.2) | 3.3 (1.4) | 3.4 (1.2) | 3.3 (1.2) | 3.6 (1.2) | 3.6 (1.2) | 3.6 (1.1) | 3.3 (1.2) |
| Pictograph (1–5) | 2.7 (1.2) | 2.6 (1.3) | 2.9 (1.3) | 2.4 (1.1) | 2.5 (1.4) | 2.5 (1.3) | 2.9 (1.3) | 2.4 (1.2) |
| Line graph (1–5) | 3.6 (1.1) | 3.3 (1.3) | 3.5 (1.1) | 3.7 (1.1) | 3.5 (1.2) | 3.3 (1.2) | 3.4 (1.3) | 3.6 (1.3) |

*MD is the mean difference from text only group.
†1=not at all easy to 10=extremely easy.
‡1= not at all satisfied to 10=I am totally satisfied.
§1=no experience to 5=a lot of experience.

prespecified equivalence threshold, suggesting the differences among the three graph groups are not clinically meaningful (see online supplemental additional file 2).

Results reporting the analysis of the association between comprehension and health literacy, education level and numeracy are provided in online supplemental additional file 2.

### Decision intention

At baseline, about two-thirds of participants (66.0% in text-only, 64.6% in bar graph, 62.7% in pictograph, 68.1% in line graph) chose the option that a child with AOM should usually take antibiotics. This increased after viewing the intervention in three of the groups to (72.8%, 79.1% and 73.6% in text-only, pictograph and

line graph, respectively) while in the bar graph group, the percentage remained similar (64.6%). However, when statistically tested, there was insufficient evidence of a difference between intervention groups for change in decision intention from preintervention to postintervention (p=0.087). Regardless of the allocated group, more people chose to give the child antibiotics after receiving the prognostic information (n=272, 650.2% before; n=302, 720.4% after). At least half of each group did not alter their choice, regardless of which intervention they received (figure 3A). The most common reason that participants gave for choosing antibiotics was their belief that antibiotics are effective. Other reasons are reported in online supplemental additional file 2.

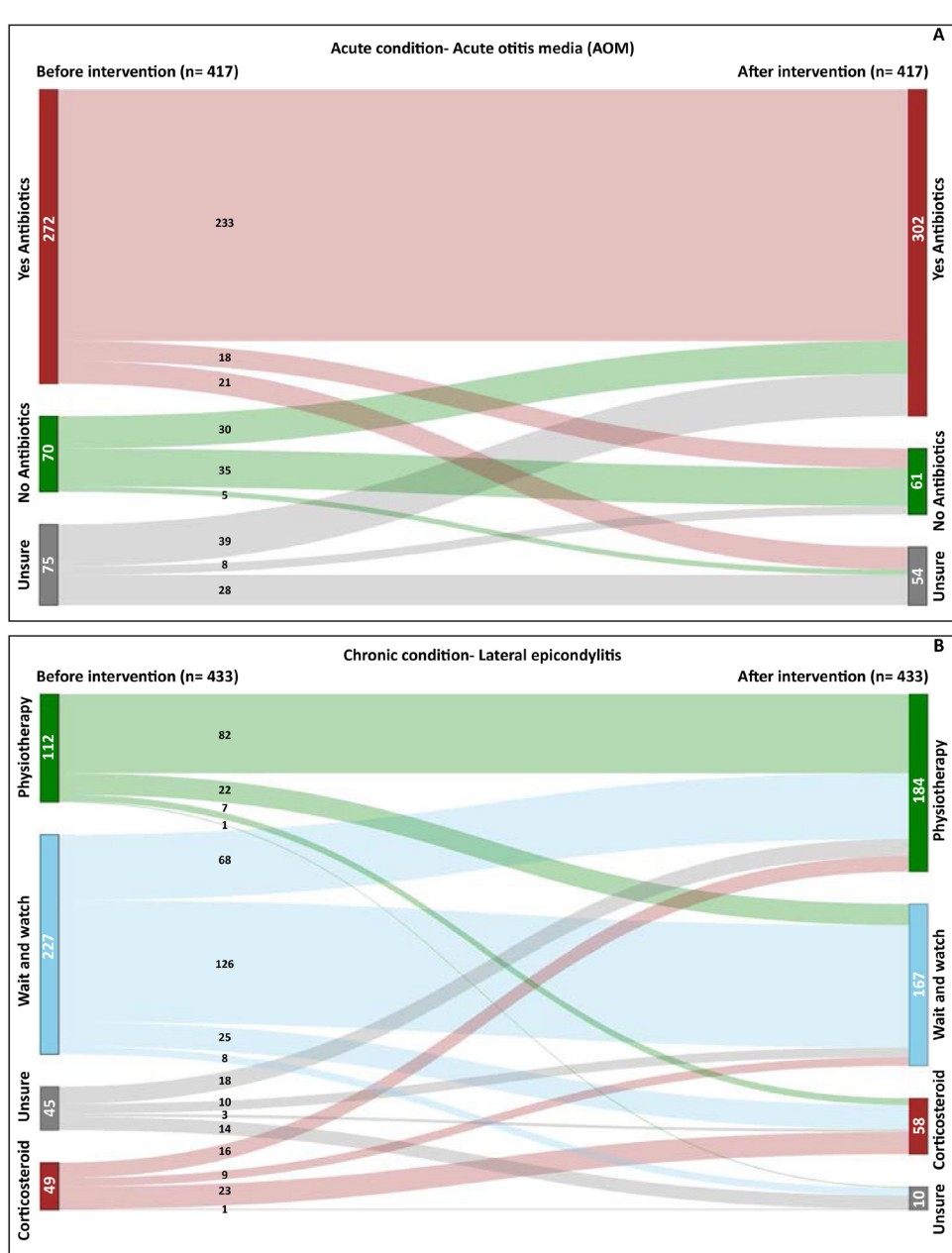

**Figure 3** Overall change in treatment decision intentions before and after viewing the interventions. The figure shows the number of participants who chose each management option in (A) acute condition—AOM and (B) chronic condition-lateral epicondylitis. AOM, acute otitis media.

### Satisfaction

Mean scores for ease of use and satisfaction were similar for all formats, with pictograph scores slightly higher (table 2). However, there was insufficient evidence of a difference between intervention groups for the how easy scale (p=0.32) and the how satisfied scale (p=0.081).

### Visual presentation preference

Participants ranked their allocated intervention as the highest, except for participants allocated to the pictograph group who ranked pictograph and bar graph equally best (30.9% of participants). Overall and regardless of the randomised group, the bar graph was the most preferred option; chosen by 137 (32.9%), followed by text-only (n=124, 29.7%). The pictograph was the option least likely to be ranked as the first choice (chosen by 58,13.9%) (table 2). For the least preferred options and reasons for preferences, see online supplemental additional file 2.

### Trial B

#### Comprehension

The difference in comprehension scores between each of the three interventions (pictograph, bar graph, line graph) and text-only was not statistically significant either before or after adjustment for age group, education level, and health literacy (table 2).

Figure 2B shows the mean comprehension score for each group. For the percentage of participants, in each group, who answered the comprehension questions correctly, see online supplemental additional file 2.

Results reporting the analysis of the association between comprehension and health literacy, education level, and numeracy are provided in online supplemental additional file 2.

#### Decision intention

Overall, before receiving any prognostic information, 52.4% (n=227) of participants intended to wait and see when asked about decision intentions. After receiving the information, 30.0% of those changed their choice to physiotherapy, while 11.0% changed to corticosteroid (figure 3B). A similar pattern was observed across all four interventions (table 2). However, when statistically tested, there was insufficient evidence of a difference between intervention groups for change in decision intention from preintervention to postintervention (p=0.88). Of the participants who chose physiotherapy (the most common choice after receiving the information, regardless of its type), the most common reasons for doing so were their understanding of the presented information, followed by their belief that this was the most effective option. For other reasons see online supplemental additional file 2.

#### Satisfaction

Mean scores for ease of use and satisfaction were similar for all formats, with pictograph scores slightly higher (table 2). However, there was insufficient evidence of a difference between intervention groups for the how easy scale (p=0.059) and the how satisfied scale (p=0.17).

### Visual presentation preference

Overall, the bar graph was the most preferred option for receiving prognostic information, chosen by 35.6% of participants, followed by 33.3% who ranked text-only as their most preferred format. The bar graph was also the highest to be chosen as a second preference (35% of participants). By intervention group, 40.2% of the text-only group, 49.6% of bar graph group, 40.5% of pictograph group and 34.31% of line graph group ranked their allocated intervention as their most preferred option to receive prognostic information in the future.

Table 2 shows the number and proportion of participants, in each group, who chose each format as their most preferred or least preferred option (other ranking choices are provided in online supplemental additional file 2).

### DISCUSSION

In our two parallel online randomised controlled trials, none of the interventions (pictograph, bar graph and line graph) were statistically significantly superior to text-only. There were no clinically meaningful differences in comprehension between the groups that viewed the bar graph, pictograph or line graph. After all visual presentations were revealed to all participants, the bar graph was the most preferred option for receiving prognostic information in the future; chosen by about one-third of participants in both trials. The type of visual presentation viewed did not appear to influence change in decision intention.

The results from our systematic review found no clear evidence to support the superiority of any particular visual presentation to communicate prognosis, with existing research mostly investigating the communication of long-term prognosis for various types of cancer.[6] In our current trials, using non-cancer conditions and shorter prognostic duration, the difference between the various graph types on comprehension or decision intention was not significant. We found some minor inconsistencies between participants' preferred graph type and their ratings of satisfaction and ease of use of the various graphs. It appears that no graph type is clearly superior at facilitating comprehension, strongly preferred by participants and viewed as the easiest to understand. Prior to the widespread use of a graph to convey prognostic information, piloting it with the target audience (ensuring a range of health literacy levels) may help to ensure appropriateness.

Several reviews and primary studies of methods of communicating treatment or screening benefits and harms have recommended adding visual presentations to textual information when communicating quantitative data.[13 22 34 35] For example, a test of three different visual presentations (text-only, fact box, visual aid) for

communicating mortality with and without screening for prostate and breast cancer found that visual aids (ie, a pictograph) generally increased comprehension by 18%.[24] However, in our trials, we found that the pictograph was superior to text-only, but only in trial A (AOM scenario). Reasons for this are unclear and might be explained by more complicated prognostic information, and more treatment options, were presented in trial B (eg, the prognostic course was more variable at the different time periods).

Visual presentations are often incorporated into patient decision aids, which are tools that can be used to support shared decision-making and contain quantitative information about the benefits and harms of options for managing a condition. The International Patient Decision Aids Standards collaboration recently reviewed the evidence about using and displaying numbers when designing patient decision aids and recommended adding a visual presentation (eg, pictographs, bar charts) to communicate quantitative information, without specific recommendation about which type to use.[34]

Line graphs have traditionally been avoided when communicating with patients as it has been assumed they are too difficult to understand,[18] however, our study did not find significantly poor comprehension in participants in the line graph group. These findings suggest line graphs are an appropriate graph that might be used to communicate prognostic information. As showing the trajectory of the condition over time is often very important, a line graph may be best suited to use when there are outcome data for more than two time points, as the equivalent information would need multiple pictographs (one at each time point) and the trajectory less clear.

Communicating quantitative information can be more challenging when multiple management options are available or multiple important outcomes. Further studies are needed to investigate how to best visually, and verbally, communicate more complex prognostic information.

There were potential limitations to these two trials; participants were members of an online survey provider and did not need to have experience with the conditions studied, and only spent about 15–20 min engaged in completing the survey. We were not able to assess participants' baseline knowledge as the comprehension questions asked for the exact numbers which we expected to be inapplicable for participants without showing them some information. The impact of prior experience, or lack of, is unclear as previous experience may have influenced decision intention, whereas no experience or personal relevance may have reduced engagement. As the questions about decision intention were hypothetical, motivation to properly read the information presented and consider the decision options may have been low. Additionally, to make a well-informed choice, patients need information about both the benefits and harms of each available treatment option. We did not present harms data as part of the information, and this may have influenced responses to questions about decision intention. Although we chose

two conditions that are typically managed in primary care settings, our findings should be interpreted with caution and might not apply to the communication of other health conditions and might only apply to a setting where participants are to interpret prognostic information by themselves, rather than when the information is presented as part of face-to-face communication.

Our trials have several strengths, including testing the visual presentations in two conditions with different trajectories and conducting the trials in parallel. Measures were used to prevent cross contamination between the trials (eg, participants of trial A were not invited to trial B, and participants of the piloting phase were not invited to participate in any of the trials).

## CONCLUSION

Our findings suggest that, if the information is clearly displayed, any of the tested visual presentations can be used to communicate quantitative prognostic information.

**Acknowledgements** We thank our colleagues at the Institute of Evidence-Based Healthcare (IEBH)-Bond University, for giving their feedback on the interventions and testing the survey during the prepiloting phase. A special thank you to the childcare teachers who pilot-tested trial A surveys and to Bond University members who pilot-tested trial B surveys. This work is dedicated to the memory of Professor Chris Del Mar, one of my PhD supervisors, who contributed fully to this work (passed away in March 2022).

**Contributors** EA, TH and CDM conceived the idea of this study. EA developed the interventions with support from TH, CDM and MB. MB helped in the refinement of pictographs intervention. EA, TH and CDM developed the survey. TH was reviewing and giving feedback on all the steps. Data collection was managed by Dynata under the direction of EA, TH. Data were analysed by EA, MJ and MB. EA drafted the main manuscript and developed all tables and figures with support and feedback from all authors. EA is responsible for the overall content as the guarantor. All authors reviewed the final version of the manuscript and approved it for submission.

**Funding** This work was supported by the Centre for Research Excellence in Minimizing Antibiotic Resistance in the Community (CRE-MARC), funded by the Australian National Health and Medical Research Council grant number (1153299).

**Competing interests** None declared.

**Patient and public involvement** Patients and/or the public were involved in the design, or conduct, or reporting, or dissemination plans of this research. Refer to the Methods section for further details.

**Patient consent for publication** Consent obtained directly from patient(s).

**Ethics approval** This study received ethical approval from Bond University Human Research Ethics Committee (EA00088). All methods in this study were performed in accordance with the Declaration of Helsinki. Informed consent was obtained from all Participants to receive online surveys by Dynata survey company. Additionally, before starting the survey, participants were provided with a formal invitation to participate in the study which included information about the study, and their right to withdraw at any time point with no negative consequences.

**Provenance and peer review** Not commissioned; externally peer reviewed.

**Data availability statement** Data are available in a public, open access repository. All data relevant to the study are included in the article or uploaded as supplementary information. Data is available in a public, open access repository. Analysed data relevant to the study are included in the article or uploaded as supplementary information.

**ORCID iDs**
Eman Abukmail http://orcid.org/0000-0002-6715-9097
Mina Bakhit http://orcid.org/0000-0002-6162-3362
Mark Jones http://orcid.org/0000-0001-6858-9710
Tammy Hoffmann http://orcid.org/0000-0001-5210-8548

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
