## [Reviewer comments · BMJ Open]

ARTICLE DETAILS

TITLE (PROVISIONAL)	Effect of different visual presentations on the public's comprehension of prognostic information using acute and chronic condition scenarios: Two online randomised controlled trials
AUTHORS	Abukmail, Eman; Bakhit, Mina; Jones, Mark; Del Mar, Chris; Hoffmann, Tammy

VERSION 1 – REVIEW

REVIEWER	Henselmans, Inge University of Amsterdam
REVIEW RETURNED	13-Nov-2022

GENERAL COMMENTS	This paper presents the results of two RCTs on the effect of different visual presentations of prognosis of two conditions commonly treated in primary care on participants' comprehension (primary outcome), treatment decision intention, satisfaction with the presentation and preference for visual presentation format (secondary outcomes). The trial design is robust, the reporting very thorough and the results clear and of significance for practice. Nevertheless, I do have some suggestions for improvement of the manuscript. Most of my questions concern the Methods of the study. I will address them in chronological order. Abstract The word 'care' is missing from 'primary care'. Introduction The Introduction is well written and clearly leads to the research questions. It did give me the impression that the authors were going to investigate how visual presentation of prognosis is helpful for use by GPs within a consultation, in addition to verbal information (see 103-104, but particularly 126-127). I was somewhat surprised to find out they were examining written information only, without any verbal communication (text only no video). Perhaps the authors could try to prevent such a surprise by re-phrasing some parts/words in the Intro or the RQ. Methods My compliments for the absolutely solid and thorough procedures. I do have some questions, some minor, but some also more critical. - What was evaluated in the pilot and what adjustments were made?- What does the MMS measure? And why is that important?- Why was the Minimal Important Difference set at -1 or +1? This is important and should be explained.
--

	- The measures section does not mention the open ended questions, the analysis paragraph does. For me, these answers do not add much to the story. The authors also do not make any reference to or interpret the results on the reasons for choosing certain treatment options in the Discussion. What do they conclude based on these reasons? - I would also leave out all analysis that do not involve the intervention conditions (so all analyses between participant characteristics and comprehension) from Methods and Results, as these do not provide an answer to the research question. In an already quite full paper, I would prefer to keep it lean and clean. Results - The authors present how treatment decisions differed before and after the interventions. Yet, they did not test if treatment decisions were different depending on the presentation of the prognosis? As decision intention is positioned as a secondary outcome, I did expect these kind of analyses. - The authors do not mention controlling for potential confounders in Trial B (as they did in trial A)? Were these adjusted analyses not performed? Why not? Discussion The authors conclude that in trial A, pictographs were superior. However, this result disappeared after adjusting for baseline differences in health literacy (higher HL in this group); and the difference was also not clinically meaningful (according to their definition). I would add these nuances here. And also later on: 'However, in our trials, we found that the pictograph was superior to text-only, but only in trial A (AOM scenario).' Perhaps your conclusions should simply be that there were no differences? 'The type of visual presentation viewed did not appear to influence change in decision intention'. Can you be sure? See my previous comments, you did not present the reader with an actual comparison on this outcome? 'and only spent about 15-20 minutes engaged in completing the survey' I am not sure that 15-20 is short? A limitation could be that these prognostic numbers were shown to people on paper without any health care provider present to explain the information. That is probably not how it will be done in primary care, and I feel that is an important potential limitation to the validity of the findings. I would even suggest to leave out the limitation of the lack of a baseline, which I find far less problematic. And replace it with this addition: the results apply to a setting where participants are to make sense of prognostic information by themselves. The authors make mention of the possibility that results are different in face-to-face communication, perhaps this limitation deserves some more prominence. Is there evidence that this type of information (graphs) is or is not better understood when communicated by a provider face-to-face? Additional files Relevant information and very well presented. I would leave out page 4 of additional file 2.
--	--

REVIEWER	Gong, Ni Jinan University
-----------------	------------------------------

REVIEW RETURNED	22-Dec-2022
-------------

GENERAL COMMENTS	I would like to thank BMJ OPEN for the opportunity to review this paper. This study discusses an interesting and important topic by assessing the effectiveness of bar, pictogram and line graphs in communicating prognosis to the public through a randomized controlled trial. As a whole this is a relatively clear and complete study, but there are a few minor issues that need to be addressed, as follows. Title : 1. This article seems to study the understanding of communication prognosis of acute and chronic diseases in healthy people, which is not clearly shown in the title, so it is suggested to add "acute and chronic diseases" in the title. Introduction. 1.It is recommended that the introduction further clarify the importance and necessity of the display research question by specifying the link between the research question "Does visual display promote patients' understanding of the information conveyed" and the healthy population. Methods Part : 1.It is necessary to explain why the healthy population was selected as the study population, what kind of association there is between the healthy public and the patients of the two trials in this study, and whether this study has taken into account the differences in the perceptions and feelings of the disease between the two populations. 2.This study was conducted online to collect data, as well as to implement the intervention, how was the accuracy and quality control of the study data ensured? Please provide further explanation. Discussion : 1. Overall the discussion section is written in a complete and organized manner, but it is recommended that the strengths and limitations of this study be placed in the last paragraph as a supplement, focusing on the analysis and discussion of the findings.
---

REVIEWER	Ali, Syed Adnan University of Karachi
REVIEW RETURNED	17-Jan-2023

GENERAL COMMENTS	I am agree with the findings and statistical analysis of this manuscript.
---

VERSION 1 – AUTHOR RESPONSE

Reviewer 1 comments	Response	Changes
	Line numbers are based on the tracked version	
This paper presents the results of two	Thank you, we appreciate your comments, please find our responses below	No changes

	RCTs on the effect of different visual presentations of prognosis of two conditions commonly treated in primary care on participants' comprehension (primary outcome), treatment decision intention, satisfaction with the presentation and preference for visual presentation format (secondary outcomes). The trial design is robust, the reporting very thorough and the results clear and of significance for practice. Nevertheless, I do have some suggestions for improvement of the manuscript. Most of my questions concern the Methods of the study. I will address		
--	--	--	--

	them in chronological order.		
1.	The word 'care' is missing from 'primary care'.	Thank you for bringing this to our attention.	The word "care" was added, abstract line 56
2.	The Introduction is well written and clearly leads to the research questions. It did give me the impression that the authors were going to investigate how visual presentation of prognosis is helpful for use by GPs within a consultation, in addition to verbal information (see 103-104, but particularly 126-127). I was somewhat surprised to find out they were examining written information only, without any verbal communication (text only no video). Perhaps the authors could try to prevent	Thank you, rewording of this section of the introduction has been done	Previous version read as follows: "The best ways of presenting prognostic information for conditions that are typically managed in primary care have rarely been explored. Some decisions in primary care, for both acute and chronic conditions, may be enhanced by having prognostic information. Prognostic information is often presented verbally and whether the addition of visually presented information can facilitate its comprehension has not been studied." Changes This paragraph now reads as follows: (please see lines 127-130) "Some decisions in primary care, for both acute and chronic conditions, may be enhanced by having prognostic information. Discussing prognostic information using visual presentations may improve patients' comprehension and facilitate decision-making that accommodates patients' values and preferences. The best ways of visually presenting prognostic information for conditions that are typically managed in primary care have rarely been explored."

	such a surprise by re-phrasing some parts/words in the Intro or the RQ.		
3.	What was evaluated in the pilot and what adjustments were made?	Lines 184-194 explain the two piloting stages. In the first stage, face-to-face piloting, the aim was to test the comprehensibility of the interventions and the clarity of the questions. No major changes were made, only the rewording of some comprehension questions. The second stage was online piloting, to inform the calculations of the sample size and test any technical issues. “test for any technical problems” was added to the section “patient and public involvement”	Previous paragraph reads as follows: “Piloting data were also used to adjust the sample size calculation and were not included in the trial results.” Changes line 196 This paragraph now reads as follows: “Piloting data were also used to adjust the sample size calculation and test for any technical problems. These data were not included in the trial results.”
4.	What does the MMS measure? And why is that important?	MMMS is an abbreviation for Medical Maximizing Minimizing Scale. It assesses patients’ preferences for aggressive versus more passive approaches to health care. The reason we measured this as people’s general attitudes towards healthcare interventions may influence their decision intentions.	Changes “Assesses patients’ preferences for aggressive versus more passive approaches to health care.” Was added to Table 1 footnote 4 and the methods section. Previous paragraph read as follows: Table 1 footnote 4 Medical Maximizer-Minimizer Scale (MMMS): medical maximizers (seek health care even for minor problems), medical minimizers: (avoid medical intervention unless it is necessary) Table 1 footnote 4 now reads as follows: Medical Maximizer-Minimizer Scale (MMMS): assesses patients’ preferences for aggressive versus more passive approaches to health care, medical maximizers (seek health care even for minor problems), medical

			minimizers: (avoid medical intervention unless it is necessary) Methods lines 220-221: Now reads as follows: Part 1: five demographic questions, a baseline decision intention question, a question on previous experience with the condition, the medical maximizer-minimizer scale (MMMS, assesses patients' preferences for aggressive versus more passive approaches to health care), and a validated subjective numeracy scale (SNS).
5.	Why was the Minimal Important Difference set at -1 or +1? This is important and should be explained.	We chose comprehension questions (explained lines 196-207) to test different comprehension skills such as reading data straight from the visual presentation, comparing numerical data, and interpreting the numbers to identify which treatment will provide more (or less) benefit. We asked 6 comprehension questions, each scored as 1 or 0. We acknowledge that setting a clinically important difference is difficult, especially for outcome measures such as comprehension. We were guided by a previous study (Woloshin 2011) that used 10% as a clinically important difference for a similar comprehension outcome measure. In our study, this would be 0.6, which was rounded to 1 as no part marks were possible in our scoring.	Changes: This sentence was added in the data analysis section: lines 254-256 "We were guided by a previous study that used 10% as a clinically important difference for a similar comprehension outcome measure."
6.	The measures section does not mention the open-ended questions, the analysis paragraph does. For me, these	Thanks, we agree that in the submitted version, it was not clear that we included open-ended questions and we have added this accordingly. We also agree that responses to these questions are not the focus of our research questions, hence we provide the results of these questions in	Changes These sentences were added: "and measured with a combination of ranking questions, Likert scales, and open-ended questions (Additional file 1)." The paragraph now reads as follows: (lines 212-213)

	answers do not add much to the story. The authors also do not make any reference to or interpret the results on the reasons for choosing certain treatment options in the Discussion. What do they conclude based on these reasons?	the supplementary material (Additional file 2). The design of patient-focused communication tools and aids to communicate prognostic information shouldn't be based solely on comprehension outcomes but also consider patients' satisfaction and preferences for how to receive information, hence why we explored if there were notable differences in these outcomes between the presentations. We do not focus on a detailed analysis of the reasons for why people reported that they would, hypothetically, choose or not choose specific treatments (that's been explored specifically in other research), but collected these data to ensure that none of the interventions consistently caused preference away from or towards a treatment option.	"Secondary outcomes were decision intention, satisfaction with the presented format (before revealing all formats), and format preference (after revealing all formats) and measured with a combination of ranking questions, Likert scales, and open-ended questions (Additional file 1)."
7.	The authors present how treatment decisions differed before and after the interventions. Yet, they did not test if treatment decisions were different depending on the presentation of the prognosis? As decision intention is positioned as a secondary outcome, I did expect these kinds of analyses.	Please see response to comment 6 above.	No change

8.	'The type of visual presentation viewed did not appear to influence change in decision intention'. Can you be sure? See my previous comments, you did not present the reader with an actual comparison on this outcome?	Please see response to comment 6 above.	No change
9.	I would also leave out all analysis that do not involve the intervention conditions (so all analyses between participant characteristics and comprehension) from Methods and Results, as these do not provide an answer to the research question. In an already quite full paper, I would prefer to keep it lean and clean.	Thanks for your suggestion.	Changes: We have removed the results of this analysis as suggested by the reviewer. Results of this analysis are only provided in the supplementary files (Additional file 2).
10.	The authors do not mention controlling for potential	Adjusted analysis was conducted for trial B similar to trial A. We apologise for not making it clear	Previous paragraph read as follows: Comprehension scores were higher in the pictograph group compared to the text-only

	confounders in Trial B (as they did in trial A)? Were these adjusted analyses not performed? Why not?		group (MD 0.49, 95% CI 0.04 to 0.93, p-value 0.032) however the difference was not statistically significant. The differences between the bar graph and text-only group and between the line graph and text-only group were not statistically significant (Table 2). Change The paragraph now reads as follows: lines 350-353 The difference in comprehension scores between each of the three interventions (pictograph, bar graph, line graph) and text-only was not statistically significant either before or after adjustment for age group, education level, and health literacy (Table 2).
11.	Discussion The authors conclude that in trial A, pictographs were superior. However, this result disappeared after adjusting for baseline differences in health literacy (higher HL in this group); and the difference was also not clinically meaningful (according to their definition). I would add these nuances here. 'However, in our trials, we found that	Thanks for your suggestion. We have amended this in the abstract, discussion, and in the conclusions to reflect our results (adjusted analysis)	In the abstract: Previous version “In both trials, the mean comprehension score was 3.7 for the text-only group, with a pictograph superior to text-only in trial A (mean difference (MD) 0.54, 95% CI 0.13- 0.95, p 0.011) but not in trial B (MD 0.49, 95% CI 0.04-0.93, p 0.032). Pairwise comparisons between the 4 presentation styles showed all were clinically equivalent (95% CIs between -1.0 and 1.0).” Current version: lines 61-71 “In both trials, the mean comprehension score was 3.7 for the text-only group. None of the visual presentations were superior to text-only. In trial A, the adjusted mean difference (MD) compared to text-only was: 0.19 (95% CI -0.16 to 0.55) for bar graph, 0.4 (0.04 to 0.76) for pictograph, and 0.06 (-0.32 to 0.44) for line graph. In trial B, the adjusted MD was: 0.1 (-0.27 to 0.47) for

	the pictograph was superior to text-only, but only in trial A (AOM scenario).¹ Perhaps your conclusions should simply be that there were no differences?		bar graph, 0.38 (0.01 to 0.74) for pictograph, and 0.1 (-0.27 to 0.48) for line graph. Pairwise comparisons between the three graphs showed all were clinically equivalent (95% CIs between -1.0 and 1.0). In both trials, the bar graph was the most preferred presentation (chosen by 32.9% of trial A participants and 35.6% in trial B).” In the discussion lines Previous version read as follows: In our two parallel online randomised controlled trials, there were no clinically meaningful differences in comprehension between the groups that viewed the bar graph, pictograph, or line graph. Only participants who were randomised to the pictograph presentation had a statistically significant better comprehension of quantitative prognostic information compared to participants who were randomised to text-only (MD 0.54, 95% CI 0.13- 0.95, p 0.011), but this only occurred in trial A (AOM scenario). Current version: lines 393-395 The paragraph now reads as: In our two parallel online randomised controlled trials, none of the interventions (pictograph, bar graph, and line graph) were statistically significantly superior to text-only. There were no clinically meaningful differences in comprehension between the groups that viewed the bar graph, pictograph, or line graph. In the conclusions
--	--	--	--

			Previous paragraph read as follows: “Although the pictograph was statistically superior to the text-only presentation in one of the two trials conducted, we did not find a clinically important difference between the types of graphs tested. Our findings suggest that, if the information is clearly displayed, any of the tested visual presentations can be used to communicate quantitative prognostic information.” Change: the first sentence was deleted Current paragraph reads as follows: Lines 476-478 “Our findings suggest that, if the information is clearly displayed, any of the tested visual presentations can be used to communicate quantitative prognostic information.”
12.	'and only spent about 15-20 minutes engaged in completing the survey' I am not sure that 15-20 is short?	We agree with the comment that completing the survey in this time may be sufficient. However, given the amount of information and the number of questions that were presented, and the processing and deliberation time and effort that some questions required, it is unclear to what extent all participants engaged genuinely with each of the questions.	
13.	A limitation could be that these prognostic numbers were shown to people on paper without any health care provider present to explain the information. That is	We agree with your comment that presenting this type of information by an interpreter/clinician might affect comprehension and other outcomes. Hence, why we acknowledge this in the limitations. We prefer to retain mentioning the lack of baseline comprehension data as a limitation. There is accumulated evidence that adding visual presentations including graphs in consultations can improve comprehension of the information and leads to better	Some rewording to accommodate the author suggestion: Previous paragraph read as follows: “Although we chose two conditions that are typically managed in primary health care settings, our findings should be interpreted with caution and might not apply to the communication of other health conditions and might be

	probably not how it will be done in primary care, and I feel that is an important potential limitation to the validity of the findings. I would even suggest to leave out the limitation of the lack of a baseline, which I find far less problematic. And replace it with this addition: the results apply to a setting where participants are to make sense of prognostic information by themselves. The authors make mention of the possibility that results are different in face-to-face communication, perhaps this limitation deserves some more prominence. Is there evidence that this type of information (graphs) is or is not better understood when	satisfaction and well-informed decisions. This is one of the most recent articles (Trevena et al. 2021)	different if the information was presented in face-to-face communication.” Current paragraph read as follows: Lines 464-467 “Although we chose two conditions that are typically managed in primary care settings, our findings should be interpreted with caution and might not apply to the communication of other health conditions and might only apply to a setting where participants are to interpret prognostic information by themselves, rather than when the information is presented as part of face-to-face communication.”
--	---	--	--

	communicated by a provider face-to-face?		
14.	Additional files Relevant information and very well presented. I would leave out page 4 of additional file 2.	Thanks. Page 4 is presenting the results of the association of comprehension with HL, numeracy, and education level. For the completeness of reporting, we feel that reporting these results is important. As this analysis is not the main focus of the study, we put it in the supplementary files (Additional file 2)	No change
	Reviewer 2 comments		
15.	This article seems to study the understanding of communication prognosis of acute and chronic diseases in healthy people, which is not clearly shown in the title, so it is suggested to add "acute and chronic diseases" in the title.	Thanks for your suggestion. We've made some changes to the title.	Previous title "Effect on comprehension of different visual presentations to communicate prognosis: Two online randomised controlled trials in healthy adults". Changes were made to the title: The new title: "Effect of different visual presentations on the public's comprehension of prognostic information using acute and chronic condition scenarios: Two online randomised controlled trials"
16.	It is recommended that the introduction further clarify the importance	Thanks for your suggestion. We did not restrict to participants who had the condition. Although, especially for the acute condition (AOM), many adults will have had this or had a family member who has had this at some stage and there would be	As per title changes above, we have removed mention of a healthy population.

	and necessity of the display research question by specifying the link between the research question "Does visual display promote patients' understanding of the information conveyed" and the healthy population.	familiarity with the condition in the sample. This is reflected in 46% in trial A and 33% in trial B reporting experience with the condition.	
17.	It is necessary to explain why the healthy population was selected as the study population, what kind of association there is between the healthy public and the patients of the two trials in this study, and whether this study has taken into account the differences in the perceptions and feelings of the disease between the two populations.	Please see response to #16. We acknowledged in our study limitations that the impact of prior experience, or lack of, is unclear as previous experience may have influenced decision intention, whereas no experience or personal relevance may have reduced engagement. Additionally, our primary outcome was comprehension of presented information, this is an objective measure and also one that is unlikely to be influenced by a person's experience, or lack of, with the condition being studied.	No change.
18.	This study was conducted online to collect data,	Many measures were put in place to ensure quality control of the data: 1- Participants were registered members of an online survey company.	Many of these measures are described in the methods section, with the others described in the

	as well as to implement the intervention, how was the accuracy and quality control of the study data ensured? Please provide further explanation.	2- Eligibility screening questions were used. If not eligible, the survey would terminate. 3- CAPTCHA check was used to ensure only humans can fill in the surveys. 4- A postcode was used to identify people who were Australian residents. If not, the survey would terminate. 5- The IP address was monitored, and the company's special unique code was used to prevent multiple entries from each registered member. 6- Participants of one trial were not invited to participate in the other trial. 7- Once participants submitted their answers, they were unable to go back and change the answers. 8- Options of multiple-choice questions were randomised to eliminate order bias in answering the questions.	supplementary (Additional file 1).
19.	Discussion : Overall, the discussion section is written in a complete and organized manner, but it is recommended that the strengths and limitations of this study be placed in the last paragraph as a supplement, focusing on the analysis and discussion of the findings.	Thanks for your suggestion.	The limitations and strengths have been moved to the last paragraph of the discussion.
	Reviewer 3 comments		
20.	I am agree with the	Thanks for your comment. We hope you enjoyed reading the manuscript.	No change

	findings and statistical analysis of this manuscript.		
--	---	--	--

VERSION 2 – REVIEW

REVIEWER	Henselmans, Inge University of Amsterdam
REVIEW RETURNED	17-Mar-2023

GENERAL COMMENTS	The authors have greatly improved the manuscript and have successfully addressed my questions and remarks. There is only one point that I feel they have not answered satisfactorily. I would appreciate if the editor would double check to see if he or she agrees with my observation, as of course it can also be a misjudgement on my part. One of the secondary outcomes is decision intention. In the revised manuscript, as in the original manuscript, the authors state that 'The type of visual presentation viewed did not appear to influence change in decision intention' (page 20) and 'In our current trials, using non-cancer conditions and shorter prognostic duration, the difference between the various graph types on comprehension or decision intention was not significant'. Yet, as far as I understand, they did not statistically test the effect of type of visual presentation on decision intention? As a matter of fact, nor did they test the differences in the other secondary outcomes (satisfaction and preferences). Their answers (point 7 and 8 in the response to the reviewer) did not address why they did not report on statistical tests for the effect of the manipulations on the secondary outcomes (yet, nevertheless draw conclusions about the existence and significance of these effects).
--

VERSION 2 – AUTHOR RESPONSE

Dear BMJ Open Editor and Reviewer

Thanks for offering the opportunity to further clarify and improve this point in our manuscript for the readers.

Please find our responses and the changes we made in blue

Reviewer's comment:

The authors have greatly improved the manuscript and have successfully addressed my questions and remarks. There is only one point that I feel they have not answered satisfactory. I would appreciate if the editor would double check to see if he or she agrees with my observation, as of course it can also be a misjudgement on my part. One of the secondary outcomes is decision intention. In the revised manuscript, as in the original manuscript, the authors state that 'The type of visual presentation viewed did not appear to influence change in decision intention' (page 20) and 'In our current trials, using non-cancer conditions and shorter prognostic duration, the difference between the various graph types on comprehension or decision intention was not significant'. Yet, as far as I understand, they did not statistically test the effect of type of visual presentation on decision intention? As a matter of fact, nor did they test the differences in the other secondary outcomes.

Response: Our secondary outcomes were descriptively reported, however, based on the editor's comment and the reviewer's suggestion we conducted statistical analyses for the secondary outcomes of decision intention and satisfaction and added the results to the manuscript. For visual presentation preference/choice, upon discussing with our team of experts, an associate professor of statistics and a professor of epidemiology, both co-authors, we have retained the reporting of preferences outcomes as descriptive, so as not to make these results unnecessarily confusing or misleading for the reader (as the between-group comparisons are complex, as each participant ranked the preferences for each format, so the number of comparison tests that would need to be conducted is very high, increasing the risk of multiplicity).

Change

Please refer to Methods section line 269-279

Methods:

"As part of the peer review process, we conducted statistical analyses for the secondary outcomes of decision intention and satisfaction. Change in decision intention from pre- to post-intervention was compared between groups using a multinomial logistic regression model with cluster robust standard errors specified to account for the pre/post repeated measures on each participant. The dependent variable was decision intention category, and the independent variables were group, time (pre/post) and the interaction between group and time. A joint test was used to test for evidence of an interaction between group and time with statistical significance set at $p < 0.05$.

ANOVA was used to test for differences in the satisfaction outcomes between intervention groups with statistical significance set to $p < 0.05$."

Results:

Trial A, decision intention (line 337-351) now reads

“At baseline, about two-thirds of participants (66.0% in text-only, 64.6% in bar graph, 62.7% in pictograph, 68.1% in line graph) chose the option that a child with AOM should usually take antibiotics. This increased after viewing the intervention in three of the groups to (72.8%, 79.1%, and 73.6%) in text-only, pictograph, and line graph respectively) while in the bar graph group, the percentage remained similar (64.6%). However, when statistically tested, there was insufficient evidence of a difference between intervention groups for change in decision intention from pre- to post-intervention ($p=0.087$). Regardless of the allocated group, more people chose to give the child antibiotics after receiving the prognostic information ($n=272,65.2\%$ before; $n=302,72.4\%$ after). At least half of each group did not alter their choice, regardless of which intervention they received. (Figure 3A). The most common reason that participants gave for choosing antibiotics was their belief that antibiotics are effective. Other reasons are reported in the supplementary files (Additional file 2).”

Trial A, satisfaction (353-356)

Mean scores for ease of use and satisfaction were similar for all formats, with pictograph scores slightly higher (Table 2). However, there was insufficient evidence of a difference between intervention groups for the how easy scale ($p=0.32$) and the how satisfied scale ($p=0.081$).

Trial B, decision intention (line 379-389)

“Overall, before receiving any prognostic information, 52.4% ($n=227$) of participants intended to wait and see when asked about decision intentions. After receiving the information, 30.0% of those changed their choice to physiotherapy, while 11.0% changed to corticosteroid (Figure 3B). A similar pattern was observed across all four interventions (Table 2). However, when statistically tested, there was insufficient evidence of a difference between intervention groups for change in decision intention from pre- to post-intervention ($p=0.88$). Of the participants who chose physiotherapy (the most common choice after receiving the information, regardless of its type), the most common reasons for doing so were their understanding of the presented information, followed by their belief that this was the most effective option. For other reasons see Additional file 2.”

Trial B, Satisfaction (line 400-403)

Mean scores for ease of use and satisfaction were similar for all formats, with pictograph scores slightly higher (Table 2). However, there was insufficient evidence of a difference between intervention groups for the how easy scale ($p=0.059$) and the how satisfied scale ($p=0.17$).

VERSION 3 – REVIEW

REVIEWER	Henselmans, Inge University of Amsterdam
REVIEW RETURNED	14-May-2023
GENERAL COMMENTS	The authors response and the adjustments in the paper are thorough and satisfactory.